# Dual Targeting Oncoproteins MYC and HIF1α Regresses Tumor Growth of Lung Cancer and Lymphoma

**DOI:** 10.3390/cancers13040694

**Published:** 2021-02-09

**Authors:** Xiaohu Huang, Yan Liu, Yin Wang, Christopher Bailey, Pan Zheng, Yang Liu

**Affiliations:** 1Division of Endocrinology, Diabetes and Nutrition, Department of Medicine, University of Maryland School of Medicine, Baltimore, MD 21201, USA; 2Division of Immunotherapy, University of Maryland Baltimore School of Medicine, Baltimore, MD 21201, USA; yanliu@ihv.umaryland.edu (Y.L.); yin.wang@ihv.umaryland.edu (Y.W.); chris.bailey@ihv.umaryland.edu (C.B.); pzheng@ihv.umaryland.edu (P.Z.); 3Department of Surgery, University of Maryland Baltimore School of Medicine, Baltimore, MD 21201, USA; 4OncoImmune, Inc., Rockville, MD 20850, USA

**Keywords:** MYC, HIF1α, protein degradation, proteasome, lung cancer, lymphoma

## Abstract

**Simple Summary:**

Both MYC and HIF1α are critical transcriptional factors involved in the initiation, transformation, progression and maintenance in a variety of human tumors, but till now no study indicates whether it is possible to simultaneously targeting both MYC and HIF1α for therapeutics. Here we report that echinomycin simultaneously inhibited MYC and HIF1α through proteasomal degradation. Treatment of echinomycin regressed tumor cell growth both in vitro cultured cells and in vivo mouse models of lung cancer and lymphoma. β-TrCP and VHL are involved in the degradation of MYC and HIF1α induced by echinomycin, respectively. Our data provided a new approach to target these and potentially other oncogenic proteins for cancer therapy.

**Abstract:**

MYC and HIF1α are among the most important oncoproteins whose pharmacologic inhibition has been challenging for the diverse mechanisms driving their abnormal expression and because of the challenge in blocking protein-DNA interactions. Surprisingly, we found that MYC and HIF1α proteins in echinomycin-treated cells were degraded through proteasome dependent pathways, respectively by the β-TrCP- or VHL-dependent mechanisms. The degradation is induced in a variety of cancer types, including those with mutations in the p53 tumor and LKB tumor suppressors and the KRAS oncogene. Consistent with inhibition of MYC and HIF1α, administration of echinomycin inhibited growth of lung adenocarcinoma xenograft and a syngeneic lymphoma model in mice. Furthermore, echinomycin efficiently induced regression of syngeneic mouse lymphoma driven by MYC over-expression. Our data demonstrated a new mechanism by which echinomycin simultaneously targets MYC and HIF1α for degradation to inhibit growth of lung cancer and lymphoma. Given the broad impact of β-TrCP or VHL in stability of oncogenic proteins, echinomycin may emerge as a non-PROTAC (proteolysis targeting chimera) degrader of oncogenic proteins.

## 1. Introduction

Cancer usually results from alterations of oncogenes and tumor suppressor genes, which are often mutated and/or abnormally expressed during tumorigenesis. Among them, MYC is one of the most potent and commonly activated oncogenes, which was mainly amplified or overexpressed in a broad range of human cancers [1]. MYC is a basic helix-loop-helix leucine zipper (bHLH-LZ) transcription factor regulating the expression of 10–15% of all genes of the genome [2,3,4]. Nuclear MYC heterodimerizes with MAX and then binds to a specific DNA sequence (E-box) to regulate different biological activities of tumor cells, including cell proliferation, growth, metabolism and metastasis [5]. MYC activation is thus considered as a hallmark of cancer initiation and maintenance [6]. Dysregulation of MYC occurs in 30–40% of human cancers, including both solid tumors and hematological malignancies, such as breast, colon, lung cancer, melanoma and myeloid leukemias [7,8]. Due to the difficulty to pharmacologically target MYC, it has been considered as an “undruggable” target in cancer therapy [9].

Hypoxia-inducible factor-1 (HIF1) is a master regulator mediating response to hypoxic stress in both normal tissues and tumors. It consists of a constitutively expressed HIF1β subunit and an oxygen-responsive HIF1α subunit, whose levels are tightly controlled. HIF1α has a very short half-life under normal oxygenated condition in normal cells [10]. Rapid degradation is mediated through von Hippel–Lindau tumor suppressor protein (VHL) which is a recognition subunit of E3 ubiquitin ligase followed by degradation through 26S proteasome [11]. In a hypoxic environment and in cells undergoing oncogenic transformation, HIF1α is stabilized [12,13] and binds to hypoxia response element (HRE) to regulate the expression of target genes. HIF1α is also stabilized in cancers under normoxia, presumably due to dysregulation of its degradation pathway. A tight association between HIF1α expression and poor prognosis and survival has been reported in different cancers including colorectal cancer, non-small-cell lung cancer (NSCLC) and pancreatic ductal adenocarcinomas patients [14,15,16]. Therefore, targeting HIF1α is of great importance to regress and inhibit the growth of tumors [17].

MYC targeting has been focused on transcriptional repression of the gene, primarily through bromine domain inhibitors [18,19]. In targeting its binding to cellular DNA, protein folding as well as translation and degradation, a number of HIF1α inhibitors have been reported [17]. Despite the intensive interests, MYC and HIF1α have not been successfully targeted for clinical cancer treatment.

Both MYC and HIF1α are nuclear transcription factors and are involved in the progress and maintenance of tumors. Recent studies revealed that dysregulated MYC can cooperate with HIF1α to promote tumorigenesis through induction of glycolytic metabolism and angiogenesis [20]. In addition, high basal MYC and HIF1α levels were observed in all multiple myeloma (MM) cell lines and primary MM cells where MYC collaborates with HIF1α to trigger VEGF production and induction of MM angiogenesis [21]. Furthermore, stabilization of HIF1α and up-regulation of MYC have been revealed in central nervous system primitive neuroectodermal tumors (CNS-PNET) animal model [22]. Therefore, concurrent targeting MYC and HIF1α may represent a better, more effective way for cancer with high expression of MYC and/or HIF1α.

In order to explore the possibility of simultaneous targeting of MYC and HIF1α, we revisited an early study which suggested that echinomycin may target transcriptional activity of both MYC and HIF1α [23,24]. Surprisingly, we observed that, rather than transcriptional repression, echinomycin induces proteasomal degradation of both MYC and HIF1α, respectively through transcriptional regulation of β-TrCP or VHL expression. Our data revealed echinomycin as a new type of oncoprotein degrader with potentially broad application for cancer therapy.

## 2. Materials and Methods

### 2.1. Cell Culture

Lung cancer cell lines (NCI-H1944, NCI-H727, Calu-1) were obtained from the American Type Culture Collection (ATCC, Gaithersburg, MD, USA). NCI-H1944, NCI-H727 and A549 were maintained in RPMI-1640 medium supplemented with 10% fetal bovine serum (FBS). Calu-1 cells were cultured in ATCC-formulated McCoy’s 5a. Eμ-Myc lymphoma cells were kindly provided by Drs. Ricky W. Johnstone and Leigh Ellis. These lymphoma cells were cultured in high glucose Dulbecco’s modified Eagle’s medium (DMEM) supplemented with 10% fetal calf serum, penicillin/streptomycin, 0.1 mM L-asparagine and 50 μM 2-mercaptoethanol in a 37 °C, 10% CO_2_ humidified incubator.

### 2.2. Cell Proliferation Assay

Cell proliferation was analyzed using Cell Proliferation Kit I (MTT) (Sigma, Burlington, MA, USA). And procedures were performed according to the instruction from manufacture. Briefly, cells were seeded at a concentration of 5 × 10^4^ cells/well in 96-well plate, and then cells were treated with vehicle or echinomycin at different concentrations for 48 h followed by adding 10 μL MTT per well. The plate was incubated at 37 °C for 4 h in a humidified incubator. 100 μL of the solubilization solution was added into each well and the plate was kept at 37 °C overnight. Then measurement was performed at 570 nm on a photo spectrometer (Synergy H4 Hybrid Reader, BioTek, Winooski, VT, USA) with the reference wavelength at 690 nm.

### 2.3. Gene Editing of vhl by CRISPR

Cultured H1944 cells in 6-well plate were infected with VHL sgRNA CRISPR All-in-One Lentivirus (Human) (Applied Biological Materials Inc., Richmond, BC, Canada ), and then infected cells were selected with puromycin (2 μg/mL). Single clones were picked and analyzed by sequencing (both genomic DNA and cDNA) and western blot. Genomic regions flanking the *vhl* gene-editing site were PCR amplified using PrimeSTAR HS DNA Polymerase (Takara, Mountain View, CA, USA) and PCR purified using a QIAquick PCR Purification Kit (QIAGEN, Hilden, Germany) followed by sequencing. For the transcript analysis, RNAs were isolated using the Trizol method (Invitrogen, Carlsbad, CA, USA) and were transcribed into cDNA using SuperScript^®^ III First-Strand Synthesis System (Thermo Fisher Scientific, Waltham, MA, USA). Then, the sequence spanning the whole vhl cDNA was amplified by PCR and purified using QIAquick PCR Purification Kit (QIAGEN) or a QIAquick Gel Extraction Kit (QIAGEN). Purified products were either sequenced directly or sequenced after being cloned into pCMV6-Entry vector through In-Fusion^®^ HD Cloning Kit (Clontech, Mountain View, CA, USA).

### 2.4. Colony Formation Assay

5 × 10^4^ cells were seeded in each well in 6-well plate. After 3 weeks in culture, the cells were then treated with vehicle or echinomycin at different concentrations for 48 h. Medium was changed with fixing/staining solution (0.05% (*w*/*v*) crystal violet, 1% formaldehyde, 1% methanol in PBS buffer). Cells were stained for 20 min at room temperature. Plate was washed by water and air dried. The colonies were either photographed or counted. The total number of colonies (>50 cells) per well was counted using a microscope.

### 2.5. Western Blotting

Whole-cell lysates were prepared in RIPA buffer (Thermo Fisher Scientific, Bedford, MA, USA) supplemented with phosphatase (Sigma Aldrich, St. Louis, MO, USA) and protease inhibitors (Roche, Brighton, MA, USA). Protein quantification was performed with the BCA Protein Assay kit (Pierce, Waltham, MA, USA) and 20 μg of protein was loaded in NuPAGE 4–12% Bis-Tris Protein Gels (Invitrogen). Then, proteins were transferred onto nitrocellulose or polyvinylidene fluoride (PVDF) membranes. After 1 h blocking at room temperature using 5% blocking milk in phosphate -buffered saline solution/0.1% Tween (PBST), membranes were incubated overnight with primary antibody in 2.5% milk in PBST at 4 °C. After the incubation, membranes were washed three times in PBST and incubated with secondary antibody for 1 h at room temperature with shaking. After three-time washing in PBST, membranes were developed using regular ECL Western Blotting Substrate or SuperSignal™ West Femto Maximum Sensitivity Substrate (Pierce) and visualized using the film processor (SRX-101A, Konica, Wayne, NJ, USA). Quantification of Western blots was done using the Image J software (NIH, Bethesda, MD, USA). For stripping, membrane was vigorously shaken in Restore™ PLUS Western Blot Stripping Buffer (Thermo Fisher Scientific) for 10 min. After stripping, membrane was washed three times in PBST followed by regular immunoblotting process. The following primary antibodies were used for western blotting: MYC (D84C12, Cell Signaling Technology); HIF1α (GTX127309, GeneTex Inc., Irvine, CA, USA); β-ACTIN (MABT825, EMD Millipore, Burlington, MA, USA).

### 2.6. Quantitative RT-PCR

RNA was isolated using the Trizol method (Invitrogen) and was transcribed into cDNA using SuperScript^®^ III First-Strand Synthesis System (Thermo Fisher Scientific) or All-In-One RT MasterMix (Applied Biological Materials Inc., Richmond, BC, Canada). Quantitative RT-PCR (qRT-PCR) was performed by real-time analysis using Power SYBR Green PCR Master Mix (Applied Biosystems, Foster City, CA, USA) or SYBR Green qPCR Master Mix (Low ROX) (Bimake, Houston, TX, USA). *Actin* and *GAPDH* were used as internal control for human derived cells and mouse derived cells, separately. Sequences of primers are shown in Appendix A.

### 2.7. Electrophoretic Mobility Shift Assay (EMSA)

LightShift™ Chemiluminescent EMSA kit (20148, Thermo Scientific) was used to assess the DNA binding ability of MYC and Hif1α. The assay was performed according to manufacturer’s protocol. Active recombinant c-Myc protein (31117) and Max protein (31244) were purchased from Active Motif (Carlsbad, CA, USA). Active c-Myc and Max protein (0.25 µg/each protein) was mixed with 50% Glycerol, 15 ng/μL Poly (dI•dC), annealed Biotin end-labeled E-box DNA (20fmol) (5′-CGGAAGCAGACCACGTGGTCTGCTTCC-3′) [25] and then incubated with echinomycin for 10 min at room temperature. The DNA-protein complexes were then separated from free probes by electrophoresis on a 6% polyacrylamide gel in 0.5× TBE, and subsequently transferred onto Biodyne B Nylon Membrane (77016, Thermo Scientific). Transferred DNA was crosslinked to membrane using UV Stratalinker 2400 (Stratagene, San Diego, CA, USA). Biotin-labeled DNA was detected by chemiluminescence according to the standard instructions provided by the manufacturer.

### 2.8. Apoptosis Assays

To quantitate apoptotic cell death, cells were collected by trypsinization, washed with phosphate buffered saline and stained using the Annexin V and PI (BD Pharmingen, San Diego, CA, USA). Fluorescence was detected by flow cytometry (BD CANTO II) and analyzed using the FlowJo software. Total events (20,000) were collected and then subsequently analyzed for the percentage of Annexin V positive cells.

### 2.9. Nuclear Fractionation

Nuclear fractionation was carried out as previously reported [26] with some modifications. To isolate chromatin, cells were resuspended in buffer A (10 mM HEPES, [pH 7.9], 10 mM KCl, 1.5 mM MgCl_2_, 0.34 M sucrose, 10% glycerol, phosphatase and protease inhibitors). Triton X-100 (0.1%) was added, and the cells were incubated for 5–8 min on ice. Nuclei were collected by low-speed centrifugation (5 min, 1300× *g*, 4 °C). The supernatant was further clarified by high-speed centrifugation (10 min, 20,000× *g*, 4 °C) to remove cell debris and insoluble aggregates, and this fraction was saved as cytosolic fraction. Nuclei were washed once in buffer A, and then lysed in buffer B (3 mM EDTA, 0.2 mM EGTA, phosphatase and protease inhibitors) for 30 min on ice. Insoluble chromatin was collected by centrifugation (5 min, 1700× *g*, 4 °C), washed once in buffer B, and centrifuged again under the same conditions. The supernatant was saved as nucleoplasmic fraction. The final chromatin pellet was resuspended in RIPA plus 0.5% SDS lysis buffer and sonicated for 60 s with 15 s intervals for four cycles in a sonicator using a microtip at 25% amplitude.

### 2.10. Protein Degradation and mRNA Stability Analysis

To check the protein degradation of MYC and HIF1α, H1944 cells were treated with echinomycin (8 nM), MG132 (5 µM), echinomycin plus MG132 or echinomycin (8 nM), cycloheximide (25 µg/mL), echinomycin plus cycloheximide for indicated time points in the figures. Subsequently, protein levels were determined via immunoblotting. For mRNA stability determination, H1944 cells were co-incubated with Actinomycin D (ActD, 5 µM) and vehicle or echinomycin for various time points as indicated in the figures. RNAs were collected using the Trizol method and quantified by qRT-PCR.

### 2.11. Xenogeneic Animal Studies

All mouse experiments were conducted in accordance with standards of animal care and approved by the Children’s National Medical Center (CNMC) Animal Care and Use Committee. Female 6- to 8-week-old Athymic NCr-nu/nu mice (Charles River Laboratories, Frederick, MD, USA) were used for animal experiments with H1944 cells-derived xenografts, and maintained under pathogen-free conditions. Cells were thawed from liquid nitrogen and recovered for 48 h. One flank of these nude mice was injected subcutaneously per mouse with recovered H1944 cells (1 × 10^6^). After inoculation, mice were weighed and tumors were measured by calliper every 3 days. Tumor volume was calculated using the following formula: tumor volume = (L × W^2^) × 0.52, in which L and W refer to the long and short tumor diameter, respectively. Mice were randomized into vehicle group and echinomycin treated group according to tumor volume. Six days after cell inoculation (diameter of tumors is around 2.5 mm), mice received 30 µL vehicle or liposome formulated echinomycin (0.14 µg/µL) by subcutaneous injection (200 µg/kg). The injection was repeated once 1 week after the first injection. No obvious toxicities were observed in the vehicle- or drug-treated animals as assessed by the change of body weight between vehicle- and drug-treated mice. All the mice in the vehicle- and echinomycin-treated groups were euthanized and tumors were taken out for the measurement of tumor weight.

### 2.12. Transfection and In Vivo Transplantation

Eμ-Myc B-cell lymphoma cells were kindly provided by Drs. Ricky W. Johnstone and Leigh Ellis. Eμ-Myc cells were cultured in high-glucose DMEM supplemented with 10% fetal bovine serum, penicillin/streptomycin, 0.1 mM L-asparagine and 50 μM 2-mercaptoethanol in a 37  °C, 10% CO_2_ humidified incubator [27]. Eμ-Myc cells were infected with firefly luciferase (FLuc)-F2A-GFP-IRES-Puro lentivirus (Biosettia, San Diego, CA, USA) and sorted using flow cytometry after selecting by puromycin.

For in vivo study, C57BL/6 male mice (6–8 weeks old) were injected (intravenous, tail vein) with 2 × 10^6^ Eµ-myc lymphoma cells expressing GFP. Echinomycin (100 µg/kg) or PBS (vehicle control) was injected (intravenous) every other day for 5 times. Whole blood (20–30 µL) was drawn from each mouse at indicated time points. Following lysis of red blood cells by ammonium-chloride-potassium (ACK) lysing buffer, GFP positive cells (lymphoma) were measured by flow cytometry (BD FACSCanto II) after staining with CD45 antibody for 20 min and analyzed using FlowJo software (version 7.6.1, Ashland, OR, USA).

### 2.13. Statistics

Statistical data are expressed as mean ± SEM and Student’s *t*-test was used to determine the significance of the difference. Kaplan–Meier analysis was performed and comparisons were done using the log-rank (Mantel–Cox) test. A *p*-value < 0.05 was considered statistically significant.

## 3. Results

### 3.1. Echinomycin Induces Degradation of MYC and HIF1α Proteins

It is assumed that HIF1α regulates gene expression by binding to core DNA sequence 5′-ACGTG-3′ within the hypoxia response element (HRE) of target gene promoters. Although echinomycin inhibits DNA-binding activity to HRE, IC_50_ for HRE binding (>40 nM) was significantly higher than EC_50_ in reporter assay [23]. These data raised the intriguing possibility that biological activities of echinomycin may be at least partially independent of blocking HIF1α binding to HRE. As shown in Appendix A, the MYC E-box motif is very similar to that of HRE. While echinomycin was known to block MYC reporter activity, its effect on MYC binding to MYC (E-box) (5′-CACGTG-3′) has not been evaluated. To fill in this gap, we performed electrophoretic mobility shift assay (EMSA) to test whether the effect of echinomycin on MYC binding to E-box explained its effect on MYC transcription activity. Appendix A showed echinomycin inhibited MYC DNA-binding activity in a dose-dependent fashion from 4 nM to 400 nM, with an IC_50_ of about 40 nM.

To measure efficacy of echinomycin on MYC or HIF1α transcriptional activity, HEK293 cells were transiently transfected with cDNA encoding MYC or mutant HIF1α (P402A/P564A) in conjunction with either E-box-driven or HRE-driven EGFP reporter, and the transfected cells were treated with vehicle or 4 nM echinomycin at the same time. At 24 h after transfection, flow cytometry analysis showed 13.7 ± 0.85% and 21.7 ± 0.92% cells were GFP^+^ respectively in MYC plus E-box-GFP or HIF1α plus HRE-GFP transfected cells. However, GFP^+^ cells decreased to 1.61 ± 0.10% and 2.25 ± 0.12% in MYC plus E-box-GFP and HIF1α plus HRE-GFP transfected cells, separately, upon echinomycin treatment (Appendix A). Therefore, EC_50_ for transcriptional activities of both HIF1α and MYC are well below 4 nM. These data raised the intriguing possibility that echinomycin may inhibit both HIF1α and MYC activities by mechanisms other than blocking their binding to DNA.

To substantiate this observation, the non-small cell lung cancer (NSCLC) cells H1944 were treated with various concentrations of echinomycin, and cell viability was assessed by MTT assay after 48 h. The concentration of echinomycin resulting in 50% cell death (IC_50_) was 3.9 ± 1.1 nM in this assay (Appendix A). In addition, we further checked the effect of echinomycin on the survival and proliferation of H1944 cells using clonogenic assay (Appendix A). A significant reduction in H1944 cell growth was observed by echinomycin treatment from 2 to 8 nM. And apoptotic cells labeled by annexin V increased after treatment by 4 nM echinomycin for 48 h (Appendix A). These data indicated that echinomycin can effectively inhibit cell growth and induce apoptosis of NSCLCs and the EC_50_ is well below IC_50_ measured by DNA-binding assays.

In our effort to identify mechanisms for echinomycin pharmacological effect in repressing MYC and HIF1α function, we found echinomycin decreased HIF1α protein in a dose dependent manner from 1 to 8 nM in H1944 cells when they were treated for 24 h (Figure 1A). We also found MYC oncoprotein showed almost the same trend as HIF1α that echinomycin effectively diminished its protein level (Figure 1A,C). Figure 1B further showed a time dependent decrease of HIF1α and MYC proteins from 0 to 24 h by echinomycin at the concentration of 4 nM.

We explored whether this regulation occurs at transcriptional or posttranscriptional levels. Our data demonstrated that mRNAs of both *myc* and *hif1α* were paradoxically increased after echinomycin treatment in dose- and time-dependent manners (Figure 1C). This is substantiated with multiple primer pairs for *myc* and *hif1α* spanning exon-exon junctions (Appendix A).

To substantiate this finding, we exploited another lung cancer cell line H727, two breast cancer cells BT474 and MDA-MB-231, lymphoma and leukemia cell lines Eμ-Myc and THP1. As shown in Appendix A, the protein contents of both MYC and HIF1α were consistently decreased upon the treatment of echinomycin for 24 h in lung and breast cancer cell lines. In addition, echinomycin decreased MYC and HIF1α protein levels in leukemia cell line THP1 (Appendix A) from hematologic malignancies as well. Therefore, echinomycin broadly inhibited MYC and HIF1α function via diminishing their protein level at higher efficacy than inhibiting their DNA binding ability. Consistent with reduction of MYC and HIF1α protein levels, the expression of MYC and HIF1α target genes, including *tert*, *brca1*, *ldha* and *glut1*, decreased upon echinomycin treatment (Figure 1D). Taken together, these data showed that, echinomycin reduced protein levels of MYC and HIF1α in spite of the increased mRNAs after treatment.

JQ1 is an inhibitor of BET bromodomain proteins and can transcriptionally inhibit the expression of MYC [18,19,28]. We tested the effect of JQ1 on H1944 cells. At IC_50_ of about 1 μM, JQ1 is much less potent than echinomycin (IC_50_ = 3.9 ± 1.1 nM), and the lung cancer cells were more resistant to JQ1 than echinomycin as approximately 40% cells are still alive even after 48 h treatment by 20 μM JQ1, while less than 20% viability was observed when the same cells received 4 nM of echinomycin (Appendix A). As expected, JQ1 transcriptionally inhibited the expression of *myc* (Appendix A) leading to reduction of MYC protein (Appendix A). In contrast to the effects of echinomycin, JQ1 had no effect on transcript but increased protein of HIF1α (Appendix A).

### 3.2. Degradation of MYC and HIF1α via Proteasome Dependent Pathways

To test if echinomycin differentially affected MYC and HIF1α proteins at different cellular localization, we compared cytosol, chromatin and nucleoplasm from vehicle and echinomycin-treated cells and evaluated the protein levels by western blot. As shown in Figure 2A, MYC was found primarily in chromatin where it was reduced by echinomycin treatment. HIF1α was found in the chromatin, nucleoplasm and cytosol and its levels at all location were reduced by echinomycin. The fact chromatin-associated MYC and HIF1α were reduced by echinomycin ruled out the possibility that inhibiting their binding to DNA contributes to their degradation.

To further characterize the degradation of chromatin-bound MYC and HIF1α induced by echinomycin, we checked the kinetics of reduction in chromatin-bound MYC and HIF1α proteins after being treated with cycloheximide (CHX), an inhibitor of protein synthesis, plus vehicle or 8 nM echinomycin. The degradation of MYC and HIF1α was faster in CHX plus echinomycin group than CHX plus vehicle group (Figure 2B). Consistent with the rapid turnover of HIF1α, its protein degradation was largely completed within 30 min after CHX treatment with the half-life t_1/2_ = 14.4 min. However, the degradation is further accelerated by 4.5 min with the half-life of 9.9 min in cells that received echinomycin plus CHX. Similarly, echinomycin promoted the degradation of the MYC protein. Treatment of echinomycin decreased the half-life (t_1/2_) of MYC from 16.1 min to 11.2 min.

To determine if echinomycin induced degradation of MYC and HIF1α is by proteasome-dependent pathway, we treated H1944 cells with 5 μM MG132, a potent proteasomal inhibitor, with or without echinomycin. Echinomycin decreased protein level of MYC and HIF1α, and MG132 caused accumulation of both MYC and HIF1α in a time-dependent manner (Figure 2C,D). However, concurrent treatment of MG132 with echinomycin abolished the effect of echinomycin on MYC and HIF1α proteins, which indicates that echinomycin induced proteasome-dependent degradation of MYC and HIF1α. Together, these data indicate that echinomycin induced degradation of MYC and HIF1α through proteasomal pathway other than through the regulation of mRNAs.

### 3.3. β-TrCP and VHL Are Involved in the Degradation of MYC and HIF1α Induced by Echinomycin, Respectively

A number of proteins have been shown to regulate stability of MYC [29]. We reasoned that critical proteins responsible for echinomycin-induced HIF1α and MYC degradation may be found among those that are regulated by echinomycin. Therefore, we tested levels of protein that had been implication in MYC degradation (Appendix A) or stabilization (Appendix A). Nuclear E3 ligases were also tested (Appendix A). Among those possible candidates, we found β-TrCP was selectively reduced by echinomycin treatment (Appendix A and Figure 3A,B). β-TrCP is an E3 ligase which can increase the stability of MYC by antagonizing SCF(Fbw7)-mediated turnover [30]. To test the role for β-TrCP in echinomycin-induced MYC degradation, we infected H1944 cells with lentivirus carrying human β-TrCP. We found MYC degradation was inhibited in β-TrCP transduced cells (Figure 3C and Appendix A). Thus, while only 25.4 ± 2.6% MYC protein remained after 4 nM echinomycin treatment in vector-transfected H1944 cells, 58.2 ± 6.1% MYC remained in β -TrCP-transfected cells that received echinomycin (Figure 3C). Therefore, β-TrCP can partially protect MYC from degradation induced by echinomycin, even though not completely. These data indicate that echinomycin induced the degradation of MYC by destabilizing MYC via decreasing β-TrCP.

Since von Hippel−Lindau protein (VHL) mediated pathway played a primary role in HIF1α degradation [31], we tested if echinomycin affected VHL levels. Strikingly, we found the VHL increased after echinomycin treatment in a both dose- and time-dependent manner (Figure 3A,B). To investigate the direct role of VHL in HIF1α degradation induced by echinomycin, we checked the response of HIF1α to echinomycin in VHL knockout (VHL-KO) H1944 cells generated by CRISPR-Cas9 genome editing technology (Appendix A). As shown in Figure 3D and Appendix A, VHL-KO cells were resistant to echinomycin-induced HIF1α degradation. These data demonstrated that VHL was responsible for the echinomycin-induced degradation of HIF1α.

To test whether these two degradation pathways are independent, we first examined the response of HIF1α to echinomycin in β-TrCP OE H1944 cells. As shown in Appendix A, in contrast to the degradation kinetics of MYC protein, the degradation of HIF1α was similar in β-TrCP OE cells when compared with control cells. Likewise, the degradation of MYC was not abrogated in VHL-KO cells (Appendix A). Therefore, the degradations of MYC and HIF1α are independently regulated.

### 3.4. Degradation of MYC and HIF1α Induced by Echinomycin Is Independent of p53 and LKB1

Major portion of human cancers have defects or mutations in the p53 tumor-suppressor pathway, which has been implicated in the regulation of MYC and HIF1α in tumor cells [32,33]. We therefore tested if the functions of echinomycin on MYC and HIF1α are p53-dependent. We found that, in p53-null cells Calu-1 lung cancer cells (Figure 4A), MYC and HIF1α were effectively degraded following treatment with echinomycin. The degradation of MYC and HIF1α showed a both dose- and time-dependent manner (Figure 4B,C). Consistent with the change of MYC and HIF1α, cell viability of Calu-1 showed a dose-dependent decrease when they were treated with echinomycin from 1nM to 8 nM (Figure 4A). Therefore, the protein degradation of MYC and HIF1α induced by echinomycin is p53-independent.

Previous study reported JQ1 induced apoptosis in NSCLC cells of different genotypes through transcriptional repression of *myc*, and that LKB1 mutation compromises sensitivity of these cells to JQ1 [34]. These data prompted us to test whether echinomycin also represses MYC by LKB1-dependent mechanisms. Since A549 cells harboring LKB1 mutation [34], they were highly resistant to the treatment of JQ1 as more than 60% cells survived even when they were treated with 20 μM JQ1 for 48 h (Figure 4D). However, echinomycin effectively inhibited the growth and viability of these cells with EC_50_ ≈ 9.5 nM (Figure 4D). Consistent with previously reported data [34], JQ1 had no effect on the expression of MYC and perhaps induced the expression of HIF1α in A549 cells (Figure 4E). In sharp contrast, both MYC and HIF1α proteins were effectively degraded by echinomycin in A549 cells (Figure 4F). These results suggested that echinomycin may be used for the treatment of a broader range of cancers than JQ1 because echinomycin is effective regardless of p53 and LKB1 status.

### 3.5. Therapeutic Effect of Echinomycin in Animal Models of Non-Small Cell Lung Cancer (NSCLC)

To further investigate the antitumor effects of MYC and HIF1α inhibition in vivo, we first explored xenograft model using H1944 cells which were originally from a non–small cell lung cancer patient. H1944 cells were injected into right flank of immunocompromised nude mice, and liposome formulated echinomycin [35] was administered subcutaneously near tumor on day 6 and day 14 at 200 µg/kg. As shown in Figure 5A, H1944 tumors grew significantly slower in echinomycin-treated mice compared with vehicle treated controls, complete elimination of tumors were observed in 4 out of 12 mice (Figure 5A,B and Appendix A). The average tumor volume was 70.93 ± 28.12 mm^3^ in echinomycin treated group 32 days after tumor cell transplantation, which was about 22% of vehicle treated mice (320.1 ± 54.33 mm^3^) (Figure 5A). Consistent with these data, echinomycin effectively reduced tumor weight when compared with vehicle (0.063 ± 0.019 g in echinomcyin treated mice vs. 0.25 ± 0.042 g in vehicle treated mice) on day 32 (Figure 5B). No gross toxicity was observed during the treatment, as there were no notable changes in body weight in echinomycin treated mice (Appendix A).

In order to confirm inhibition of MYC and HIF1α can inhibit the growth of lung cancer, we used mouse *Kras^G12D/+^; p53^−/−^* lung cancer cell line [36] for further validation. These cells also showed good response to echinomycin treatment at nanomolar concentration (Appendix A). These cells had low expression of HIF1α, but with high expression of MYC (Appendix A). It is striking that, when treated with 4 nM echinomycin for 48 h, MYC is almost gone (Appendix A). When the tumor cells were transplanted into C57BL/6 via intravenous (IV) injection, they quickly formed metastatic lesions in the lung. Echinomycin remarkably inhibited the growth of tumors after three treatments by IV injection at the concentration of 250 µg/kg (Figure 5C,D). In vehicle-treated group, tumor cells formed lesions throughout the lung, while echinomycin-treated group significantly retarded the growth (Figure 5E).

### 3.6. Echinomycin Inhibits Growth of Eμ-Myc Lymphoma

Most aggressive B-cell lymphomas, such as Burkitt lymphoma, are characterized by dysregulation of the *myc* gene. We next tried to address if echinomycin could be applied for the therapy of lymphoma using Eμ-Myc mouse model of human Burkitt lymphoma. Eμ-Myc lymphoma cells which were originally from Eμ-Myc transgenic mouse showed high sensitivity to echinomycin, with an IC_50_ of approximately 0.25 nM (Figure 6A). Echinomycin down-regulated MYC and HIF1α proteins despite increasing their mRNA levels in Eμ-Myc lymphoma cells (Appendix A).

To determine the efficacy of echinomycin on preventing tumor progression, mice were inoculated with GFP-expressing Eµ-myc lymphoma cells and treated with echinomycin or vehicle control. Mice transplanted intravenously with lymphoma rapidly developed lymphomas and typically succumbed to the disease 3–4 weeks after transplantation. However, treatment of these mice with echinomycin resulted in a dramatic reduction of the tumor burden, as measured by bioluminescence in vivo (Figure 6B,C). We quantified the GFP expressing cells among peripheral blood leukocytes by flow cytometry. An average of 23.12 ± 4.58% GFP^+^ cells in vehicle group and 7.31 ± 1.59% in echinomycin treated mice were documented by flow cytometry (Figure 6D,E). Vehicle treated mice showed splenomegaly and enlarged liver, while echinomycin to some extent rescued these changes (Appendix A). Notably, we observed significantly less tumor nodules in liver in echinomycin treated mice than that in control group (Figure 6F–I). All these tumor bearing mice died within 22 days after transplantation in vehicle group, at which time point one third mice were still alive in echinomycin treated group, although all of them died within 26 days after transplantation. With these overall therapeutic benefits by echinomycin, drug treated mice showed prolonged survival when compared with control mice (Figure 6J).

## 4. Discussion

MYC and HIF1α are critical in human cancer pathogenesis through regulation of metabolism, angiogenesis, cell proliferation and apoptosis, and self-renewal of cancer stem cells. Here we showed that echinomycin may be used to target their degradation for the purpose of cancer therapy. Our work presented herein makes three seminal points.

First, echinomycin causes degradation rather than blocking transcriptional activities of these oncoproteins. Previous result already showed echinomycin inhibited HIF1α binding to the hypoxia-responsive element (HRE) DNA sequence of its target genes, and here we provided direct experimental evidence that echinomycin also inhibited DNA binding ability of MYC to E-box. However, the dose required to inhibit DNA binding is at least 10-fold higher than their ability to inhibit MYC and HIF1α function. These data prompted us to explore mechanism independent of inhibiting DNA-protein interaction. While previous studies have shown that echinomycin can reduce HIF1α protein levels [37], it was not explored whether this was achieved through synthesis or degradation of HIF1α. Here, we addressed this question from the perspective of both RNA and protein levels. Our results clearly showed the mRNA of *myc* and *hif1α* paradoxically increased while MYC and HIF1α proteins decreased. Since the decrease is reversed by proteasome inhibitor, it must be achieved by proteasomal pathway. We further showed that echinomycin caused degradation of HIF1α and MYC in a variety of cancer types tested here, including lung cancer, breast cancer, lymphoma and leukemia. These data suggest broad application of the drug in cancer therapy. A recent report showed echinomycin reduced the *myc* RNA in NB4 and Jurkat cells [38], which suggests that echinomycin may suppress MYC function by multiple mechanisms.

For traditionally non-druggable cancer targets, it is now desirable to target them for degradation through PROTAC [39]. However, such an approach requires structure-based design of chimera compound to binds target protein of as well as E3 ligase. Despite the strong interest, no PROTAC is available for either MYC or HIF1α to our knowledge. Here, we empirically identified echinomycin as a non-PROTAC-based degrader for HIF1α and MYC. Given the broad over-expression and biological significance of these oncogenes, echinomycin may emerge as an important tool for cancer therapy.

Second, echinomycin-induced degradation of HIF1α and MYC is independent of p53 and LKB tumor suppressors. Recent studies suggested p53 is involved in the regulation of MYC and HIF1α [32,33] and many anti-cancer drugs work in a p53-dependent manner. In the current study, our results demonstrated echinomycin effectively induced the degradation of MYC and HIF1α and inhibited the growth of p53 deficient lung cancer cells, which suggests the function of echinomycin on MYC and HIF1α is p53-independent and it can be used for the treatment of both p53 wild-type and mutated cancers. *Myc* can be transcriptionally regulated by BET bromodomain protein 4 (BRD4), and BET bromodomain inhibitor JQ1 selectively downregulates MYC and MYC-dependent target genes and showed anti-tumor effects of hematological malignancies and NSCLC [18,19]. However, some tumors are resistant to the treatment of JQ1, which has subsequently been revealed by the loss of LKB1 [34]. In contrast, echinomycin promoted the degradation of MYC regardless of LKB1 status. In addition, JQ1 can only be used for the treatment of tumors whose MYC expression is transcriptionally activated. However, echinomycin can be used for the tumors with highly expressed MYC as a result of enhanced transcription and/or protein stability. In addition, our data demonstrated that although JQ1 transcriptionally inhibited *myc*, it stabilized HIF1α protein without affecting its RNA level, which may limit its application to HIF driven solid tumors and blood malignancies. This may also alternatively explain why some tumor cells are resistant to the treatment of JQ1. In contrast, echinomycin can effectively induce the degradation of MYC and HIF1α, and inhibit their functions. Therefore, these functional features of echinomycin raised the intriguing possibility that echinomycin may be tested for a broad range of cancers regardless of the status of p53 and LKB1, and how MYC is over expressed.

Third, we showed that echinomycin induced HIF1α degradation by causing over-expression of VHL. Since VHL is also involved in degradation of MDM2 [40] among other oncoproteins, it is possible that echinomycin can degrade MDM2. Likewise, degradation of MYC is hereby shown related to inhibition of β-TrCP expression, which was known to stabilize YAP [40] in addition to MYC. It would be of interest to test potential impact of echinomycin on the YAP oncoprotein, which represents another major undruggable oncoprotein [41].

## 5. Conclusions

In summary, our data demonstrated that MYC and HIF1α protein were degraded through proteasome dependent pathways, by the β-TrCP- and VHL-dependent mechanisms, respectively, in echinomycin-treated cancer cells. In addition, administration of echinomycin inhibited growth of both lung adenocarcinoma xenograft and a syngeneic lymphoma model in mice. By showing that echinomycin simultaneously causing proteasomal degradation of MYC and HIF1α, our data provided a new approach to target these and potentially other oncogenic proteins for cancer therapy.

## Figures and Tables

**Figure 1 cancers-13-00694-f001:**
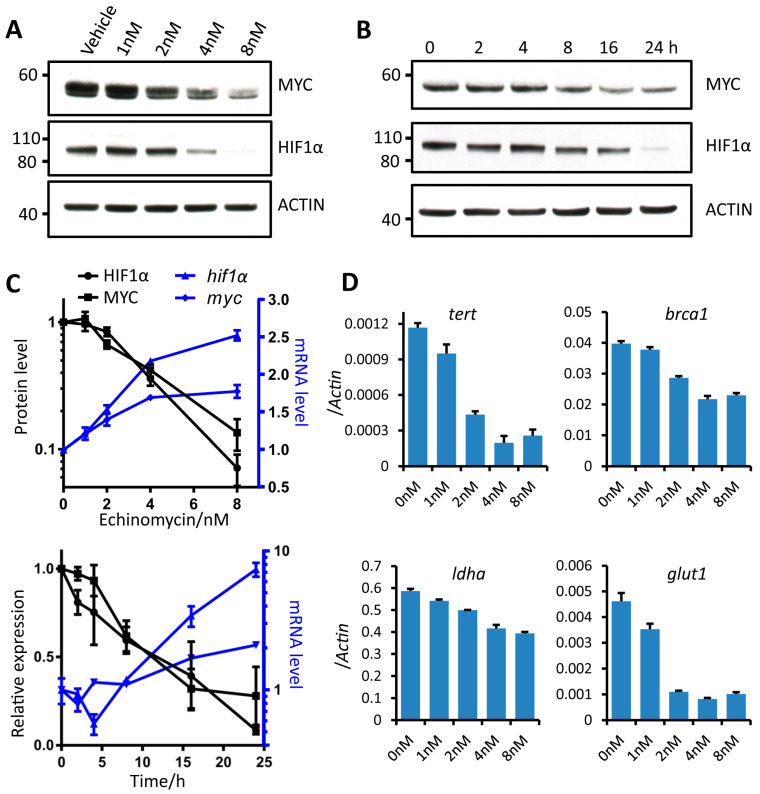
Echinomycin reduced MYC and HIF1α proteins but increased their mRNAs. (**A**) Western blot analysis of MYC and HIF1α proteins in H1944 lung adenocarcinoma cells treated with echinomycin at indicated concentrations for 24 h. Experiments were repeated 4 times. The uncropped Western blots have been shown in Appendix A. (**B**) Time-dependent decrease of MYC and HIF1α proteins in H1944 cells treated with 4 nM echinomycin at indicated time points. Experiments were repeated 3 times. (**C**) Quantification of MYC and HIF1α protein levels (*n* = 4 replicates) and *myc* and *hif1α* RNAs in H1944 lung adenocarcinoma cells treated for 24 h with echinomycin at indicated concentrations (upper panel) (*n* = 3 replicates) or treated with 4 nM echinomycin at indicated time points (lower panel) (*n* = 2 replicates). Experiments were repeated 3 times. (**D**) Expression of MYC and HIF1α target genes in H1944 cells treated with echinomycin at indicated concentrations for 24 h by qRT-PCR (*n* = 3 replicates). Experiments were repeated 2 times.

**Figure 2 cancers-13-00694-f002:**
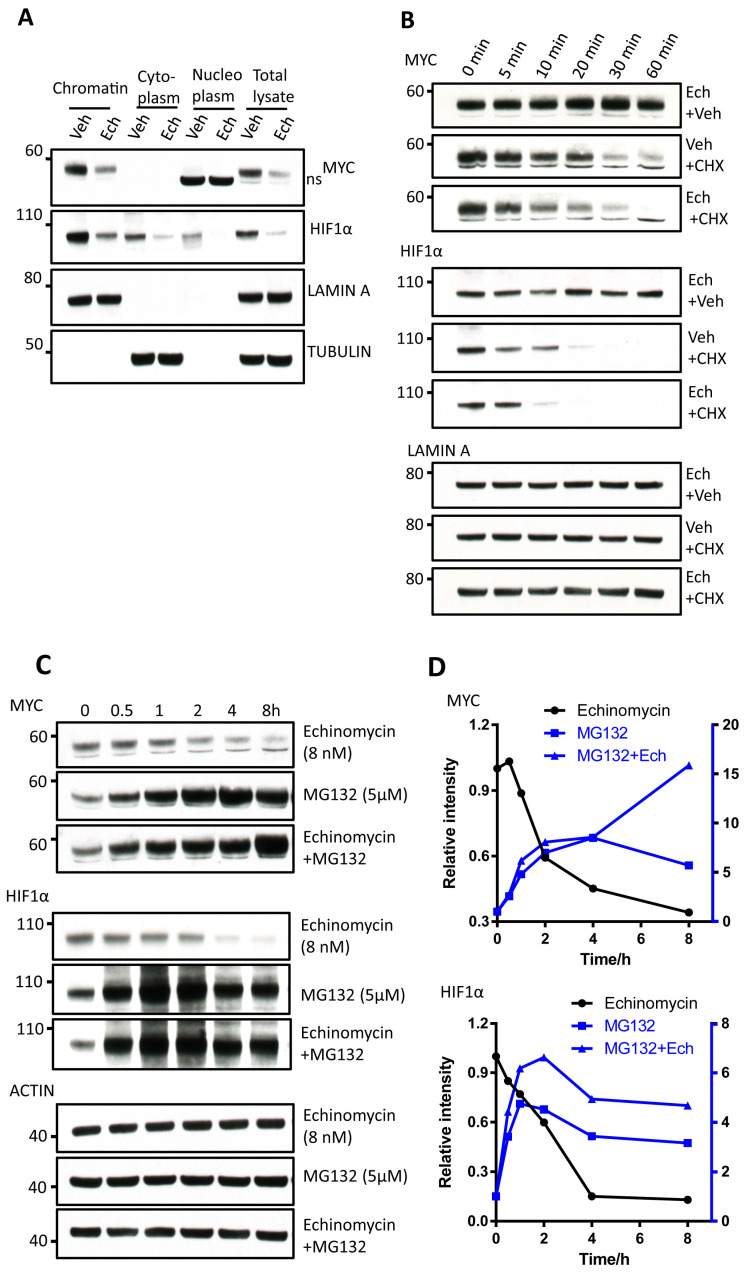
Induction of proteasomal degradation of MYC and HIF1α by echinomycin. (**A**) MYC and HIF1α in cytosol, nucleoplasm and chromatin in H1944 cells treated with 4 nM echinomycin (Ech) or vehicle (Veh) for 24 h were analyzed by western blot after fractionation. Experiments were repeated 3 times. (**B**) Chromatin-bound MYC and HIF1α in H1944 cells treated with echinomycin (Ech, 8 nM), cycloheximide (CHX, 25 µg/mL) or Ech + CHX for indicated time were checked by western blot. Experiments were repeated two times. (**C**) Quantification of MYC and HIF1α in (**B**). (**D**) H1944 cells were treated with echinomycin (Ech, 8 nM), MG132 (5 µM) or Ech + MG132 for indicated time, and MYC and HIF1α proteins were analyzed by western blot. Experiments were repeated two times. Quantification of MYC and HIF1α in (**D**).

**Figure 3 cancers-13-00694-f003:**
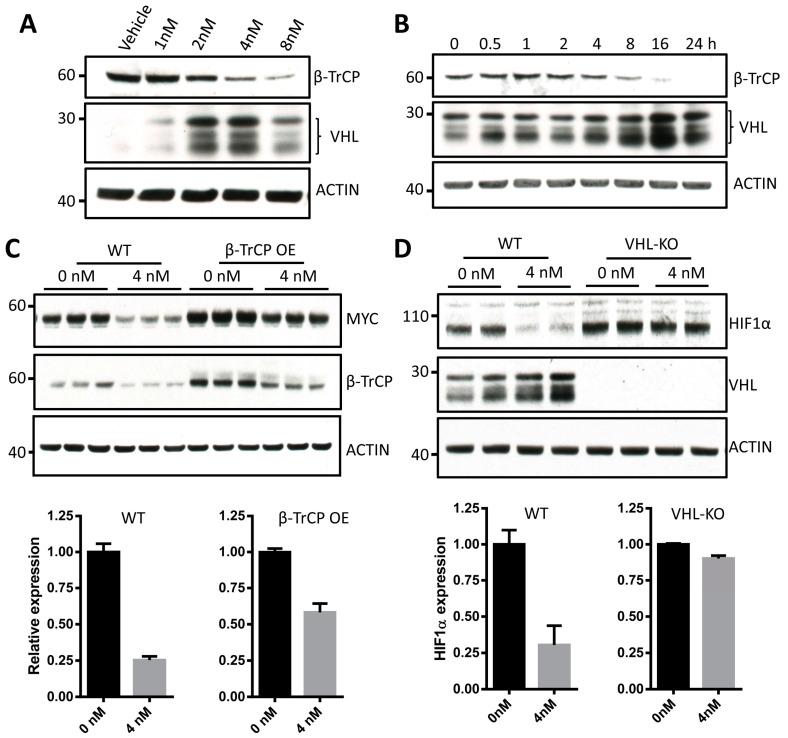
β-TrCP and VHL are involved in the degradation of MYC and HIF1α induced by echinomycin, respectively. (**A**) Western blot analysis of β-TrCP and VHL proteins in H1944 lung adenocarcinoma cells treated with echinomycin at indicated concentrations for 24 h. (**B**) Time-dependent response of β-TrCP and VHL proteins in H1944 cells treated with 4 nM echinomycin at indicated time points. (**C**) Western blot and quantitative analysis of MYC in wild type H1944 cells and β-TrCP transduced H1944 cells treated with vehicle or 4 nM echinomycin for 24 h. (**D**) Western blot and quantitative analysis of HIF1α in wild type H1944 cells and VHL deficient H1944 cells treated with vehicle or 4 nM echinomycin for 24 h. Experiments were repeated 2–3 times.

**Figure 4 cancers-13-00694-f004:**
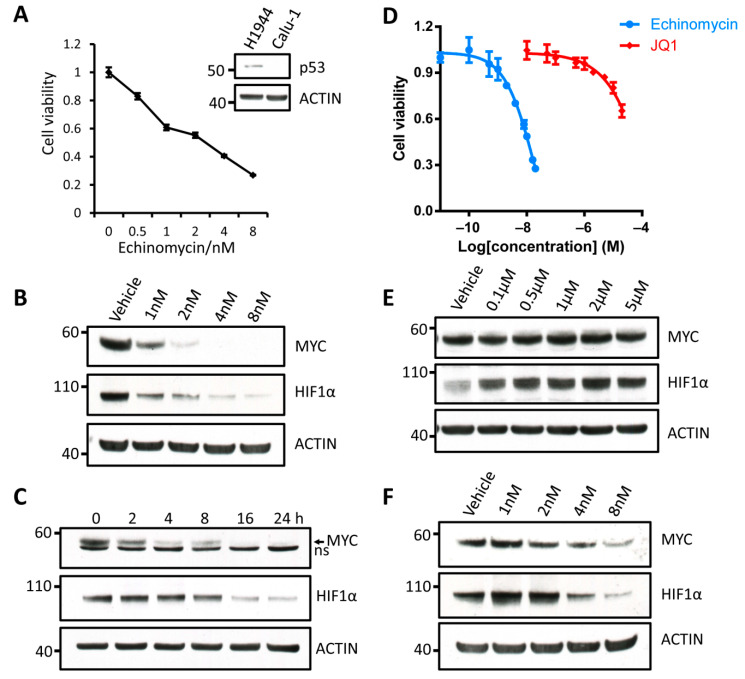
Degradation of MYC and HIF1α induced by echinomycin is p53 and LKB1 independent. (**A**) Cell viability and proliferation assays by MTT in Calu-1 cells. Calu-1 cells were treated with echinomycin for 48 h, and then cell viability and proliferation were checked by incubating with MTT overnight (*n* = 3 replicates). Insert shows the expression of p53 in H1944 and Calu-1 by western blot. (**B**,**C**) Protein levels of MYC and HIF1α were analyzed in Calu-1 cells treated with different concentrations of echinomycin for 24 h (**B**) or treated with 4 nM echinomycin for indicated time (**C**). ns: non-specific band. Experiments were repeated three times. (**D**) MTT assay in A549 lung cancer cells. A549 cells were treated with either echinomycin or JQ1 for 48 h at indicated concentrations, and then cell viability was determined by MTT assay (*n* = 3 replicates). (**E**) Protein levels of MYC and HIF1α in A549 cells were checked by western blot when treated with JQ1 at different concentrations. (**F**) MYC and HIF1α proteins were checked by western blot in A549 cells which were treated by echinomycin for 24 h at indicated concentrations. Experiments were repeated two times.

**Figure 5 cancers-13-00694-f005:**
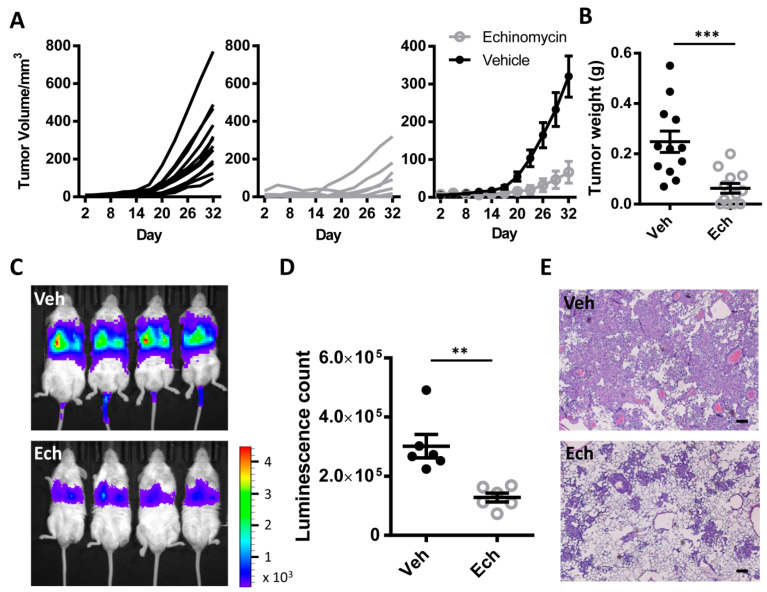
Targeting MYC and HIF1α inhibited tumor growth in lung adenocarcinoma in xenograft mouse models. (**A**) Tumor growth of H1944 xenograft. H1944 cells (1 × 10^6^) were injected s.c. into nude mice and treated with vehicle or echinomycin (200 µg/kg) on day 6 and day 13 after cell transplantation. Tumor volumes were determined by caliper measurements. Left panel: tumor volume of vehicle treated mice; Middle panel: tumor volume of echinomycin treated mice; Right panel: average tumor volume of vehicle and echinomycin treated mice. n = 12 mice/group. (**B**) Tumor weight was measured on day 32 after cell transplantation in vehicle and echinomycin treated H1944 xenograft mice. *n* = 12 mice/group. ***, *p* = 0.003. (**C**) Representative images show the tumor cells in mice that received transplants of luciferase-expressing mouse *Kras^G12D/+^; p53^−/−^* lung cancer cells in vehicle and echinomycin treated groups. (**D**) Quantification of bioluminescence count in (**C**). *n* = 6 mice/group. **, *p* = 0.002. (**E**) H & E staining shows the distribution of *Kras^G12D/+^; p53^−/−^* cancer cells in vehicle and echinomycin treated mouse lung tissues. Scale bar: 200 µm.

**Figure 6 cancers-13-00694-f006:**
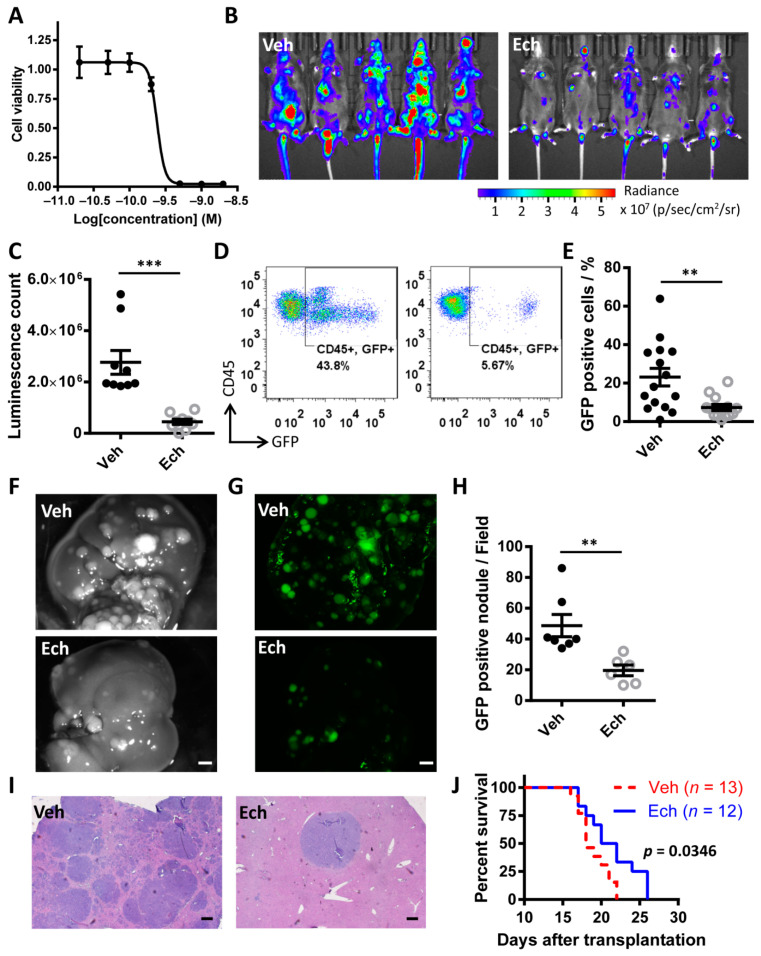
Inhibition of MYC and HIF1α exhibited anticancer activity in Eμ-Myc B-cell lymphoma transplanted mouse model.(**A)** Growth inhibition curve of Eμ-Myc cells treated with echinomycin for 48 h at indicated concentrations (*n* = 4). (**B**) In vivo real-time bioluminescence imaging of luciferase-expressing Eμ-Myc lymphoma cells in vehicle and echinomycin treated mice. (**C**) Quantification of bioluminescence count in (**B**). *n* = 9 mice/group. ***, *p* = 0.0006. (**D**,**E**) Representative images (**D**) and quantification results (**E**) showing GFP positive cells (lymphoma) in the blood in vehicle and echinomycin treated mice by flow cytometry. *n* = 15 mice in Veh and 13 in Ech. **, *p* = 0.005. Experiments were repeated two times. Morphology of tumor nodules in liver were imaged under bright field (**F**) and fluorescence (**G**) in vehicle and echinomycin treated mice with Eμ-Myc cells transplant. Scale bar: 100 µm. (**H**) Quantification of GFP positive nodules in liver from (**G**). *n* = 7 mice in Veh and 6 in Ech. **, *p* = 0.001. (**I**) H & E staining shows the distribution and morphology of tumor nodules in vehicle and echinomycin treated mouse liver tissues. Scale bar: 200 µm. (**J**) Kaplan Meier survival curve of Eμ-Myc lymphoma bearing mice treated with vehicle or echinomycin. *n* = 13 mice in Veh and 12 in Ech. *p* = 0.0346 by log-rank (Mantel–Cox) test.

## Data Availability

The data presented in this study are available on request from the corresponding author.

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
