# Peer review of "Dual Targeting Oncoproteins MYC and HIF1α Regresses Tumor Growth of Lung Cancer and Lymphoma"

_cancers, 2021, doi:10.3390/cancers13040694_

Round 1

Reviewer 1 Report

In the current study, the authors identified in vitro and in vivo anti-cancer activities of echinomycin on different types of tumor cells, which are through dual targeting oncoproteins MYC and HIF1α. They also further determined molecular mechanisms of echinomycin caused protein degradation of MYC and HIF1α. In general, this is a well-written research article with solid evidence to support their conclusion. However, there are a few weaknesses in the data presented which need to be addressed:

Major comments:

1) One major concern is that they do not examine the effects of echinomycin on cancer cells under hypoxia condition. The cell culture they showed is under normal oxygenated condition.

2) In the H1944 NSCLC xenograft model, can they explain why echinomycin was administered subcutaneously instead of i.p. injection? Do they think i.p. injection will be less effective?

3) In Figure 5, they should detect and compare the expressional levels of MYC and HIF1α between vehicle- and echinomycin-treated mice tumor tissues, by using IHC (at least in one animal model).

4) In Figure 6I, they should also detect and compare the expressional levels of MYC and HIF1α in tumor tissues using IHC.

Minor comments:

1) In Figure 1C lower panel, they should add Protein level curves just like upper panel.

2) In Figure S5B and C, they described “JQ1 had no effect on transcript and protein of hif1α”, however, it looks JQ1 increased HIF1α expression in both experiments.

3) There are a few spelling or typing errors in the text and figure legend, which should be carefully checked and corrected.

Author Response

We appreciate the reviewer's comments and please check the detail information in the attachment.

Reviewer 2 Report

The authors provide compelling and surprising data showing that echinomycin induces the proteasomal degradation of MYC and HIF. The study is a strong contribution to the literature as these are important oncoproteins. The manuscript is well written and the data are presented clearly and are of high quality. Some minor issues are mentioend below.

1) In the initial reporter assay (Figure S1) can you exclude th possibility that ther are differences in transfection efficiency? There seems to be no re4adout to ensure normalized transfection ,and could the drug have simply affected transfection or expression of the MYC/HIF transgenes?

2) When I see so many proteins degraded after drug treatment - it leads to concerns about overall cell death as a cause rather than selective effects. The authors do actually provide data (S6) showing that many important proteins are in fact stable in echinmycin treated cells. Perhaps the authors can refer to that early in the MS to draw attention to the fact that this the effect on MYC and HIF are not simply due to the cells dying? Also, showing that the cells are otherwise doing well at the time that the proteins are degraded would be helpful.

3)Are there any ill effects of echinomycin on mouse health? Do they lose weight?

Author Response

(The authors gave the same response as above.)

Reviewer 3 Report

In this manuscript, Huang and coworkers demonstrated that echinomycin inhibits proliferation of multiple cancer cell types via proteasomal degradation of MYC and HIF1α protein levels. They also found that MYC protein degradation was controlled by a mechanism involving β-TRCP while VHL regulated echinomycin-induced degradation of HIF1α. They supported their findings with xenograft lung adenocarcinoma and syngeneic lymphoma mouse models. Based on their findings, the authors suggest that the echinomycin may be useful for cancer therapy.

These studies are important especially with respect to the potential therapeutic applications. However, there are several concerns that the authors need to address in order to make their results and conclusions more valid.

  • The authors test their findings in multiple and different types of cancer cell lines however, they did not compare the effect of echinomycin with normal, non-cancer cells. The studies should be extended to normal cells to make sure that echinomycin targets only cancer cells and doesn’t have unwanted effects on other cell types. This would also evaluate echinomycin therapeutic potential.

  • A major limitation of this data is the lack of insight on the mechanism of echinomycin induced MYC and HIF1α degradation. Although the authors showed that degradation pathways of MYC and HIF1α are regulated independently, they did not clarify the relationships of MYC with β-TRCP and HIF1α with VHL. The authors conclude, based on their findings, that echinomycin induced the degradation of MYC by decreasing β-TRCP, and VHL was responsible for the echinomycin-induced degradation of HIF-1α. However, they didn’t confirm their statement experimentally. Do the members of these two pathways influence cell proliferation through their physical interaction? How does echinomycin affect it?

  • The authors conclude that β-TRCP can protect MYC from degradation induced by echinomycin. However, in Fig 3C, β-TRCP OE elevates the levels of VHL correspondingly in both untreated and echinomycin treated cells, indicating that the β-TRCP OE increases VHL protein levels independently of echinomycin treatment. The authors need to address these results.

  • Western blots for MYC in Figures 4B and 4C were performed on the same Calu-1 cell line. What could be the reason for the strong unspecific band in Fig. 4C?

  • Fig 4F - it is not clear which cell line was used for this experiment. The authors talk about H1944 cells in the text but indicate A549 cells in the description of Fig 4F.

Author Response

(The authors gave the same response as above.)

Reviewer 4 Report

  In this article, the authors evaluated the anti-tumor effect of echinomycin on many cancer. The author carefully provides a lot of very strong evidence to prove the anticancer effect of echinomycin in vivo and in vitro. In addition, they found that the mechanisms of anti-cancer effects are mediated by the degradation of Myc and HIF-1alpha, respectively. The novelty and importance are impressive. However, there are still a few questions and suggestions to the authors.

  1. There are some typos, such as β-. TrCP is on page 9 and the cells transfected with -TrCP are on line 9 of page 10. Please check the text again.
  2. Are the cancer cells present in the tumor nodules more resistant to echinomycin after treatment with echinomycin?

  3. In Figure 6D, why did the lymphoma cells in the blood of mice treated with echinomycin express high levels of GFP, while the lymphoma cells in the blood of the control group mice expressed varying levels of GFP?

  4. Figure 6J shows the survival curve of Eμ-myc lymphoma-bearing mice treated with vehicle or echinomycin. Although the P value of the result is significant, the prolonged survival time is not as long as I expected based on the results of in vitro experiments. Did the death of these mice treated with echinomycin due to tumor or other reasons, such as loss of tumor immunity or infection? It seems insufficient to assess the toxicity of echinomycin by weight change alone. The author can analyze the proportion and activity of immune cells in the future (at least T lymphocytes are important in tumor immunology).

  5. What is the change in the number of cancer stem cells after echinomycin treatment?

Author Response

(The authors gave the same response as above.)

Round 2

Reviewer 1 Report

I think the authors have addressed (or provide reasonable explanation) all of comments.

Reviewer 3 Report

At this moment, I believe that publication of the revised version of the manuscript will interest Cancer readers.